# Psychometric Properties of the Berger HIV Stigma Scale: A Systematic Review

**DOI:** 10.3390/ijerph182413074

**Published:** 2021-12-11

**Authors:** Stanley W. Wanjala, Ezra K. Too, Stanley Luchters, Amina Abubakar

**Affiliations:** 1Department of Public Health and Primary Care, Campus UZ-Ghent, Ghent University, 9000 Ghent, Belgium; 2Department of Social Sciences, Pwani University, Kilifi P.O. Box 195-80108, Kenya; 3Institute for Human Development, Aga Khan University, Nairobi P.O. Box 30270-00100, Kenya; ezra.too@aku.edu (E.K.T.); amina.abubakar@aku.edu (A.A.); 4Department of Epidemiology and Preventive Medicine, Monash University, Melbourne, VIC 3004, Australia; 5Department of Public Health, Pwani University, Kilifi P.O. Box 195-80108, Kenya; 6Department of Psychiatry, University of Oxford, Oxford OX3 7JX, UK; 7Neuroassessment Group, KEMRI/Wellcome Trust Research Programme, Centre for Geographic Medicine Research (Coast), Kilifi P.O. Box 230-80108, Kenya

**Keywords:** HIV/AIDS, psychometrics, stigma, HIV stigma scale

## Abstract

Addressing HIV-related stigma requires the use of psychometrically sound measures. However, despite the Berger HIV stigma scale (HSS) being among the most widely used measures for assessing HIV-related stigma, no study has systematically summarised its psychometric properties. This review investigated the psychometric properties of the HSS. A systematic review of articles published between 2001 and August 2021 was undertaken (CRD42020220305) following the Preferred Reporting Items for Systematic Reviews and Meta-Analyses (PRISMA) guidelines. Additionally, we searched the grey literature and screened the reference lists of the included studies. Of the total 1241 studies that were screened, 166 were included in the review, of which 24 were development and/or validation studies. The rest were observational or experimental studies. All the studies except two reported some aspect of the scale’s reliability. The reported internal consistency ranged from acceptable to excellent (Cronbach’s alpha ≥ 0.70) in 93.2% of the studies. Only eight studies reported test–retest reliability, and the reported reliability was adequate, except for one study. Only 36 studies assessed and established the HSS’s validity. The HSS appears to be a reliable and valid measure of HIV-related stigma. However, the validity evidence came from only 36 studies, most of which were conducted in North America and Europe. Consequently, more validation work is necessary for more precise insights.

## 1. Introduction

The HIV/AIDS pandemic continues to be a major public health burden, affecting millions of people globally, with significant morbidity and mortality being reported. According to UNAIDS, approximately 37.7 million people were living with HIV/AIDS globally in 2020 [1]. Furthermore, there were approximately 680,000 HIV/AIDS-related deaths across the globe in 2020 [1]. This puts HIV/AIDS among the top 20 leading causes of death globally [2].

HIV-related stigma remains a significant impediment to the eradication of the HIV/AIDS pandemic. Across the globe, HIV-related stigma has been a contributing factor in delays in HIV testing [3,4] and engagement with HIV care [5]. Moreover, among people living with HIV (PLWH), it has had a role in suboptimal adherence to antiretroviral therapy (ART) [6] as well as disengagement from HIV care [7,8]. This has led to poor outcomes such as non-viral suppression [9] and faster infection progression [10]. Furthermore, HIV-related stigma has been associated with negative consequences that may further impede progress towards eradicating the pandemic, such as non-disclosure of HIV-positive status [11] and poor mental health functioning [9].

To adequately address HIV-related stigma, it needs to be measured using appropriate measurement tools [12]. Several tools have been developed for this purpose. These include the Internalized AIDS-related Stigma Scale [13], the Stigma and Social Impact Scale [14], the T.B. and HIV/AIDS Stigma Scale [15], the HIV Stigma Scale developed by Sowell and colleagues [16], the HIV Stigma Scales developed by Visser and colleagues [12], the HIV/AIDS Stigma Instrument [17], the Stigma mechanisms of the HIV stigma framework [18], the Internalized HIV Stigma Measure [19], and Berger’s HIV Stigma Scale (HSS) [20], among others. To quantify the burden of HIV-related stigma adequately and accurately, these tools should be psychometrically sound across the diverse population of PLWH from different settings. Psychometric properties describe a scale’s reliability and validity for use in a given population [21].

One of the most commonly used tools is the HSS developed by Berger and colleagues [20]. This tool is a 40-item measure that assesses perceived stigma in PLWH using a four-point Likert scale (strongly disagree = 1, disagree = 2, agree = 3, strongly agree = 4). The scale consists of four subscales that assess the various mechanisms through which PLWH experience stigma: personalised stigma, disclosure concerns, negative self-image, and concern with public attitudes. The personalised stigma subscale assesses the perceived consequences of other people knowing about an individual’s HIV status. The disclosure concerns subscale assesses an individual’s concerns or worries about disclosing their HIV status. The negative self-image subscale assesses an individual’s negative feelings towards oneself due to HIV. Finally, the concern with public attitudes subscale assesses people’s attitudes towards people with HIV.

During its development, the scale was shown to have excellent reliability and validity [20]. The scale, including its subscales and abbreviated versions, has since been validated and used widely among different HIV-positive populations in different settings, such as among the youth in Thailand [22], children in Sweden [23], men who have sex with men (MSM) in the United States [24], and women in Indonesia [25]. In these studies, it was observed that the scale was reliable and/or valid for use in these diverse sub-populations and settings.

Despite its extensive use and evidence of adequate psychometric properties across different settings, data on the scale’s psychometric robustness has not been systematically summarised. For meaningful and accurate data that can inform the development and evaluation of HIV-stigma reduction interventions, researchers and related practitioners involved in HIV research and care who intend to use this scale require information on its psychometric robustness. Therefore, to address the above-mentioned gap, this study aims to systematically summarise the available data on the psychometric properties of the HSS in terms of reliability, content and face validity, construct validity, convergent and divergent validity, discriminant validity, and cross-cultural adaptation.

## 2. Materials and Methods

### 2.1. Protocol and Registration

This study’s protocol was developed and registered in the International Prospective Register of Systematic Reviews (PROSPERO) under registration number CRD42020220305. This study followed the Preferred Reporting Items for Systematic Reviews and Meta-Analyses (PRISMA) guidelines [26].

### 2.2. Search Strategy

We performed a comprehensive bibliographic search on PubMed, Web of Science—core collection, PsycINFO, Scopus, and Embase (Excerpta Medica dataBASE) for relevant articles that were published from 2001 (when the scale was first published) to 24 August 2021 (when the last search was conducted).

Our search structure included keywords such as “Stigma”, “HIV infections”, “Scale”, and “Berger” combined by the Boolean operator “AND”. Respective synonyms for these keywords were joined using the “OR” Boolean operator. Where applicable, Medical Subject Headings (MeSH) terms were used. The search strategy was adapted to fit the specifications of the different databases. Appendix A provides the search string used in the PubMed database.

The search was limited to articles published in the English language where a database could allow this filter. All the identified references were retrieved and uploaded to the EPPI Reviewer Web software (https://eppi.ioe.ac.uk/EPPIReviewer-Web/Main, accessed on 21 September 2021) for data management. Additionally, we manually searched the reference lists of the included studies for additional relevant articles. We also searched the Open Grey database for potential grey literature that met our inclusion criteria.

### 2.3. Eligibility Criteria

#### Inclusion and Exclusion Criteria

To be included in the review, studies had to fulfil pre-determined inclusion and exclusion criteria. We included studies where the HSS (including abbreviated versions and subscales) was being developed and/or validated for use among PLWH or in the HIV-negative but affected population. We also included any studies (observational or experimental) that used any version of the HSS to assess HIV-related stigma and reported a psychometric measure of reliability and/or validity.

We excluded studies that used the HSS but did not report psychometric properties. We also excluded studies that used the HSS and reported the psychometric properties of the original scale or an earlier version of the scale. Studies that adapted the HSS for use in a population other than PLWH or HIV-affected participants were excluded. Studies that constructed scales using a mix of items from the HSS and other scales were also excluded. Studies published in languages other than English, qualitative studies, reviews, and studies for which the full text could not be found were all excluded. For duplicate reports from the same project, only the main and more comprehensive paper was considered.

### 2.4. Screening of Articles by Inclusion and Exclusion Criteria

For potential inclusion, all the identified articles from the database search were independently screened by two reviewers (S.W.W. and E.K.T.) in two steps: (i) by title and abstract and (ii) by full text. The reviewers held a meeting at the end of every step to resolve disagreements. Disagreements between the reviewers were consistently low and were resolved through consensus.

### 2.5. Data Extraction

Data extraction was conducted in the EPPI-Reviewer Web software by S.W.W. and E.K.T., who shared the included studies equally. The following information was extracted from the included studies: (i) article details—the name of first author, title, and year of publication; (ii) study information—country, study design, study setting, sampling method, and source of the sample; (iii) sample characteristics—the population involved, sample size, age (mean, median, or range), and sex (proportion of females); (iv) characteristics of the scale used—version of the scale used, the number of items in the scale, and mode of administration; (v) outcome—the reported psychometric properties of the scales used.

### 2.6. Quality Assessment

Two reviewers (S.W.W. and E.K.T.) independently assessed the quality of the included studies using the COnsensus-based Standards for the selection of health Measurement INstruments (COSMIN) checklist [27,28,29]. The two reviewers then resolved any disagreements in the quality rating through consensus. The COSMIN checklist contains standards (in terms of design requirements and preferred statistical methods) that assess the methodological quality of studies on measurement properties. The checklist contains boxes that contain the quality standards for each specific measurement property. The quality of each included study is evaluated by rating the quality standards of individual measurement properties on a four-point rating scale (inadequate, doubtful, adequate, very good) [30]. The overall rating of each study’s quality is determined by taking the lowest rating of any standard in a box (“worse score counts” principle) [30].

In this review, we only assessed the quality of the development and/or validation studies. Since the primary aim of the included observational or experimental studies was not scale development and/or validation, they inadvertently did not provide standards that would enable quality assessment using the COSMIN checklist. Therefore, we anticipated that these studies would be rated poorly even though they may be of good quality.

### 2.7. Data Analysis

The included studies were heterogeneous in terms of the versions of the scale used and the sup-populations involved. Therefore, we summarised the data narratively by the reported psychometric properties. We categorised the countries where the included studies were conducted into their respective continents and used descriptive statistics (frequencies and percentages) to summarise their distribution. Descriptive statistics were also used to summarise the years of publication, the versions of the scale used, the populations involved, and the study types.

## 3. Results

### 3.1. Results of Database Search

The initial electronic search yielded 1241 records from the different databases (Embase, *n* = 94; Pubmed, *n* = 87; PsycInfo, *n* = 162; Scopus, *n* = 845 and Web of Science, *n* = 53). A search of the reference lists of the included articles yielded 14 additional articles. We also searched the Open Grey database for grey literature but did not find any relevant articles. After removing duplicates and screening the articles by the eligibility criteria, 166 articles were included in the review [6,20,22,23,24,25,31,32,33,34,35,36,37,38,39,40,41,42,43,44,45,46,47,48,49,50,51,52,53,54,55,56,57,58,59,60,61,62,63,64,65,66,67,68,69,70,71,72,73,74,75,76,77,78,79,80,81,82,83,84,85,86,87,88,89,90,91,92,93,94,95,96,97,98,99,100,101,102,103,104,105,106,107,108,109,110,111,112,113,114,115,116,117,118,119,120,121,122,123,124,125,126,127,128,129,130,131,132,133,134,135,136,137,138,139,140,141,142,143,144,145,146,147,148,149,150,151,152,153,154,155,156,157,158,159,160,161,162,163,164,165,166,167,168,169,170,171,172,173,174,175,176,177,178,179,180,181,182,183,184,185,186,187,188,189,190]. Figure 1 shows the PRISMA flowchart for the systematic review process.

### 3.2. General Characteristics of the Included Studies

Appendix A presents the characteristics of the 166 included articles in detail. In summary, the included studies were published between 2001 and 2021, with the majority (91.6%, *n* = 152) being published from 2010 onwards. Most of the included studies (46.4%, *n* = 77) were conducted in North America (Appendix A). The remaining studies were distributed across Asia (22.9%, *n* = 38), Africa (14.5%, *n* = 24), Europe (6.6%, *n* = 11), South America (6.0%, *n* = 10), and Oceania (1.2%, *n* = 2). Four studies (2.4%) were multicountry studies that were conducted in either four [82], three [146], or two countries [99,124]. Figure 2 shows the map of the geographical distribution of these studies.

Twenty-four of the included studies [20,22,23,39,40,45,63,69,70,91,92,93,95,97,107,111,118,121,144,146,147,171,178,184] were development and/or validation studies of the HSS and/or its abbreviated versions. The remaining studies were observational or experimental studies that used the HSS to assess HIV-related stigma in different contexts.

The included studies recruited a total of 68,933 participants, with individual sample sizes ranging from 14 in Rwanda [98] to 2987 in China [187]. The recruited participants included diverse samples of PLWH, including adolescents and young adults living with HIV in 12 studies [22,23,38,62,72,80,83,125,126,129,163,178], adolescent girls and young women living with HIV in four studies [33,54,68,117], pregnant women living with HIV in one study [31], older women living with HIV in one study [151], older adults living with HIV in seven studies [44,61,63,77,81,139,161], women living with HIV in 19 studies [25,42,50,57,58,59,89,90,110,123,135,136,138,141,157,158,168,169,179], men living with HIV in six studies [46,75,78,108,153,159], MSM living with HIV in 15 studies [24,53,79,85,86,102,104,116,124,142,148,160,171,174,182], female sex workers living with HIV in three studies [51,99,190], transgender women living with HIV in two studies [34,101], inmates living with HIV in one study [114], injectable drug users in one study [39], sexually active PLWH in one study [74], and homeless PLWH in one study [177]. Four studies [40,55,106,118] recruited HIV-affected participants and adapted the HSS to assess stigma-by-association in this population. The remaining studies (*n* = 88) recruited participants from the general population of PLWH.

In terms of the versions of the HSS used, 51 studies used or validated the full 40-item version of the scale (Appendix A). Three studies [22,69,91] validated the full scale and further developed an abbreviated version of the scale. The remaining studies (*n* = 112) used study-specific abbreviated versions or subscales of the HSS to assess HIV-related stigma across various contexts.

### 3.3. Reliability of the HSS

Table 1 and Appendix A present the reliability of the HSS as reported by the development and/or validation studies and observational or experimental studies, respectively.

In summary, all but two [144,146] of the included studies reported an aspect of reliability of the HSS.

#### Internal Consistency and Test–Retest Reliability

Of the 164 studies that reported an aspect of reliability, all the studies reported the internal consistency of the HSS, except one study [166], which only reported the test–retest reliability of a 40-item version used in a US sample of PLWH. Of these, seven studies [20,39,52,88,91,92,121] additionally reported the scale’s test–retest reliability. In one study [39], split-half reliability was additionally assessed and reported to be adequate (0.93). No study reported the intra- or interrater reliability of the scale.

For internal consistency, all the studies, except a 4-item scale in one study [127] and some subscales in ten studies [23,52,68,69,91,111,170,172,176,189], reported Cronbach’s alphas of above the recommended acceptable threshold of 0.70 [191]. Two studies [34,95] reported the internal consistency of the HSS using McDonald’s omega instead of Cronbach’s alpha and reported good reliability (Table 1 and Appendix A). In one study [111], Cronbach’s alpha (0.83), ordinal alpha (0.88), and omega alpha (0.93) were reported as measures of internal consistency. For test–retest reliability, the intraclass correlations in all the studies were above the recommended cut-off score of 0.40 [191], except for the concern with public attitudes subscale in the 17-item HSS used in Puerto Rico, which reported an intraclass correlation coefficient of 0.27 [92].

### 3.4. Validity of the HSS

Table 1 and Appendix A present the validity of the HSS as reported by the development and/or validation studies and observational or experimental studies, respectively, in detail. In total, 36 of the included studies assessed an aspect of the validity of the HSS (Table 1 and Appendix A).

#### 3.4.1. Content and Face Validity

Fourteen studies [20,35,40,63,70,91,97,104,107,111,118,137,147,184] evaluated the content validity of the HSS. Of these, three studies [35,70,91] also evaluated the scale’s face validity. In three of these studies [35,104,184], content validity was assessed using content validity index, with reported content validity index of 1.00 [184], 0.82 [35], and 0.87 [104]. For the rest of the studies, the items in the scales used were judged to be relevant, comprehensive, clear, or comprehensible by the participants [40,63,70,91,107,118], experts [20,147], and participants with experts [97,111]. In one study [137], the items were selected from a previously published and validated scale to ensure content validity. To assess face validity, item relevance was judged by the participants or the experts in two studies [70,91]. One study [35] assessed face validity using the face validity index and reported a face validity index of 0.56.

#### 3.4.2. Construct Validity

In total, 27 studies assessed the factor structure of the HSS [20,22,23,40,45,60,69,70,71,91,92,93,95,97,107,111,121,143,147,156,171,176,178,180,184,190]. During the development of the initial scale, Berger et al. [20] derived a four-factor solution from the exploratory factor analysis (EFA) that accounted for 46% of the variance. The four-factor structure was replicated in 14 of these studies [45,60,69,70,71,91,92,107,111,147,156,171,178,184], often with acceptable-to-good model fit statistics (see Table 1 and Appendix A). In two studies that assessed construct validity in two versions of the HSS [22,97], the 12- and 10-item versions of the scale in these studies also replicated the four-factor structure of the original scale. However, EFA suggested a five-factor solution in the 40-item version of the scale in a study by Rongkavilit et al. [22] and a three-factor solution in the 13-item version of the scale in a study by Kamitani et al. [97]. Factor analysis for the remaining studies suggested a three-factor solution in two studies [23,176], a two-factor solution in six studies [40,93,95,121,177,180], and a one-factor solution in two studies [143,190].

#### 3.4.3. Convergent and Divergent Validity

Twenty studies [20,23,39,40,45,60,63,69,70,91,92,93,95,97,107,121,133,171,178,184] assessed the convergent validity of the HSS. Of these, one study [97] additionally evaluated the divergent validity of the scale. To assess convergent validity, these studies explored the associations of the HSS with related measures of depression [20,39,40,60,63,69,91,92,93,95,121,133,171,178,184], anxiety [40,178], self-esteem [20,39,45,95,133,171], social support [20,60,70,178], social integration [20], social conflict [20], bullying victimization [40], peer problems [40], stigmatization [45,92], discrimination [45], fear of discovery [45], quality of life [23,69], physical, psychological, and emotional wellbeing [70,107], life satisfaction [70], self-efficacy [70], disclosure of HIV status [70], sexual abuse [92], overall health status [97], acculturation [97], alcohol use [178], sexuality problems [60], perceived side effects [60], adherence [121], viral load [121], CD4 count levels [93,121], HIV symptoms [93], spirituality/religiousness [93], and sociodemographic indicators (previous incarceration, gender, sexual orientation, consistency of condom use, and satisfaction with housing) [121].

The observed associations between the HSS and these measures in these studies were as hypothesised (see Table 1 and Appendix A for correlation coefficients). Overall, there were low to high positive correlations between the scale and measures of depression, anxiety, social conflict, peer problems, bullying victimisation, stigmatisation, discrimination, fear of discovery, sexual abuse, sexuality problems, perceived side effects, HIV symptoms, detectable viral load, previous incarceration, female gender, heterosexual orientation, inconsistent condom use, and alcohol use.

Similarly, but in the inverse direction, there were low to high negative correlations between the scale and measures of self-esteem, social support, social integration, quality of life, life satisfaction, physical, psychological, and emotional well-being, self-efficacy, adherence, CD4 count level, disclosure of HIV status, spirituality/religiousness, overall health status, and acculturation. To assess divergent validity, Kamitani et al. [97] explored the associations of the 10- and 13-item versions of the HSS with education level in a sample of Asians living with HIV in the US and found no association (B = −0.28, *p* = 0.93).

#### 3.4.4. Discriminant Validity

Only four studies [40,70,111,171] assessed the discriminant validity of the HSS. First, Boyes et al. [40] evaluated the discriminant validity of a 10-item scale adapted for measuring stigma-by-association among South African youth affected by HIV by comparing its performance among HIV-affected and unaffected youth. As expected, stigma scores were significantly higher among HIV-affected youth (*p* < 0.001).

Second, in a Spanish sample of PLWH, Fuster-RuizdeApodaca et al. [70] evaluated the discriminant validity of a 30-item scale by comparing its performance among participants with and without a history of AIDS-related opportunistic infection. As expected, stigma scores were significantly higher among participants with a history of AIDS-related opportunistic infection (*p* = 0.003).

Third, using a 21-item Spanish version of the scale in Mexico, Valle et al. [171] demonstrated discriminant validity of the scale by finding significant differences between participants with lower and higher stigma scores on the scale (*p* < 0.01). Finally, Luz et al. [111] determined the discriminant validity of a 12-item scale in a Brazilian sample of PLWH by comparing stigma scores based on antiretroviral treatment and adherence status. As expected, in specific samples or across the entire sample, participants who were not on treatment or those with poor adherence showed statistically significant higher subscale scores.

#### 3.4.5. Cross-Cultural Adaptation and/or Validity

Three studies [111,144,146] assessed the cross-cultural validity of the HSS by assessing the differential item functioning (DIF) of its items across cultural and ethnic groups. In a 40-item version of the HSS, Rao et al. [144] found 11 items that functioned differently across a sample of black and white American participants. Reinius et al. [146] assessed the DIF of a 32-item version of the scale across Indian, Swedish, and US cohorts of PLWH. This study found nine items that functioned differently between cultures and one item that functioned differently across gender. Luz et al. [111] assessed the DIF of a 12-item version of the scale in a diverse sample of Brazilian participants recruited from different social media platforms (Grindr, Hornet, and WhatsApp/Facebook). They found one item that functioned differently across the three samples.

The cross-cultural adaptation process of the HSS was conducted in 34 studies (Table 1 and Appendix A). In 23 of these studies [36,39,40,69,70,91,92,94,95,97,107,111,118,121,125,126,129,132,137,156,159,162,184], the scales were adapted to the local contexts following forward and back-translations and review by experts and/or feedback from participant interviews and discussions. Ten of the studies [22,82,98,103,104,105,118,127,136,164] just translated the scale to the target native languages, while one study [122] adapted a scale that had already been translated in the study setting. Nine studies [23,35,61,88,116,147,160,170,175] did not adapt the scale but used previously adapted versions.

### 3.5. Quality of the Included Studies

Appendix A presents the quality ratings of the included development and/or validation studies per the reported psychometric measures using the COSMIN checklist. Three studies [92,144,146] were rated to be of very good quality, eight studies [22,45,69,93,95,121,178,184] to be of adequate quality, nine studies [20,40,70,91,97,111,118,147,171] to be of doubtful quality, and four studies [23,39,63,107] to be of inadequate quality.

## 4. Discussion

This review aimed to systematically summarise the available evidence on the psychometric properties of the Berger Stigma Scale. One hundred and sixty-six studies met our inclusion criteria and were reviewed. Most of the included studies (46.4%) were conducted in North America, particularly in the United States. This is not surprising, considering that the scale was first developed and validated for use in this setting. Therefore, a reliable and valid tool for use in this setting was available to researchers. The lack of adequately validated tools for use in a given setting can contribute to the under investigation or underreporting of patient-reported health outcomes. For instance, the paucity of validated screening tools for anxiety disorders has previously been cited as a reason for the lower screening of anxiety disorders among young people living with HIV from sub-Saharan Africa [192].

Most of the included studies (91.6%) were published from 2010 onwards. This highlights an encouraging increase in the scientific knowledge of HIV-related stigma over the past decade. It is also encouraging to note that the included studies recruited diverse samples of PLWH, including key populations such as MSM, female sex workers, and adolescent girls and young women. This exemplifies the efforts that have been put in place to end HIV/AIDS, including efforts towards understanding HIV-related stigma in such key populations that have been known to be significant contributors to the pandemic [193,194].

### 4.1. Psychometric Properties of the HSS

The internal consistency of the HSS ranged from acceptable to excellent (Cronbach’s alpha ≥ 0.70 [191]) in 93.2% of the included studies. This suggests that the scale is largely reliable for use across contexts and the diverse population of HIV-positive and negative but affected individuals. Moreover, for the few studies (*n* = 8) [20,39,52,88,91,92,121,166] that reported the test–retest reliability of the scale, the reported intraclass correlation coefficients were above acceptable (≥0.40 [191]) in seven of these studies. This suggests that the HSS is also stable over time and provides further evidence of the scale’s reliability. However, since this was only assessed in a few studies, future studies also need to incorporate this aspect of reliability to confidently ascertain the temporal stability of this scale.

It appears that the four-factor structure of the HSS is stable across contexts and the diverse population of PLWH. Of the 26 studies that assessed the factor structure of the HSS besides the original study, 16 studies (61.5%) replicated the four-factor structure of the original scale. Although different factor-solutions were suggested in some studies, the suggested solutions were consistent with the number of subscales used [143,177,180]. Where different factor solutions were suggested for the full or abbreviated versions, some factors in the suggested solutions were consistent with factors on the original scale. While this suggests that the experiences of HIV-related stigma might be similar across contexts, the differences in factor solutions might be due to cultural, linguistic, and sociodemographic differences [23,176] and differences in the number of items used [121].

Across studies reporting the convergent validity of the HSS, the correlations between the scale and the theoretically related measures were all in the expected direction, providing evidence of convergent validity. Of note, the correlations between the HSS and these measures were not only in the expected direction but also significant in most of the included studies. The observed associations are consistent with findings from a systematic review that assessed the association between HIV-related stigma and various health outcomes [195].

The discriminant validity of the HSS was established in four studies [40,70,111,171]. Although limited to only a few studies, the HSS appears to have the ability to distinguish between people with a higher and lower risk of HIV-related stigma. This suggests that the scale is sensitive to HIV-related stigma and provides strong evidence of validity [196]. However, four studies indicate a very thin evidence base, thus the need for more research focusing on correlating the HSS with biomarkers of HIV, e.g., medical adherence, viral load, and CD4 count. Additionally, the HSS could be used in longitudinal studies to try and see to what extent it is sensitive to change since this is what will make it useful for interventions.

Although the HSS has been translated and/or adapted to target settings, evidence of cross-cultural validity remains limited. Only three of the included studies [111,144,146] assessed and established the cross-cultural validity of the HSS through DIF. More cross-cultural validation work of the HSS is needed to determine its performance across different cultures and contexts. Despite this, the scale appears to have content and face validity across contexts, although this is limited as it was only assessed in 14 of the included studies. Overall, the evidence of reliability and validity (although limited) from this review suggests that the HSS is psychometrically sound to assess HIV-related stigma across contexts and the diverse population of HIV-positive and negative but affected people.

### 4.2. Limitations of the Review

This review was not without limitations. First, we restricted our search to studies that were only published in the English language. Consequently, we may have left out relevant studies published in different languages. Second, we could not rate the quality of the included observational or experimental studies because of the nature of our chosen rating tool. Because of this, it is difficult to ascertain the strength of evidence of the results from these studies. Finally, we could not conduct a meta-analysis because of the heterogeneity in the versions of the HSS used as well as the sub-populations involved. Because we could not conduct a meta-analysis, we could not also assess and report on publication bias.

### 4.3. Implications of the Findings to Practice and Future Research

There is a need for more validation work to ascertain the validity of the HSS across contexts. Only 21.7% (*n* = 36) of the included studies reported an aspect of the validity of the HSS, while the remaining studies only reported reliability. This validation work is needed even more in African, Asian, Oceania, and South American settings, where there is minimal information on the validity of the HSS. Of the 36 studies reporting on an aspect of validity, more than half of these studies (58.3%, *n* = 21) came from North America and Europe. The rest were distributed across Africa [40,133,137], Asia [22,35,39,91,95,104,143,180,184], and South America [69,111,121]. No study was from Oceania. The distribution of the validation studies reflects a bias towards the global North. This may be because most of the funding for research is in the global North [197] although the highest disease burden is in low- and middle-income countries [1]. This highlights a need for more research investments in low- and middle-income countries. That said, where reported, the HSS appears to have good psychometric properties across contexts and across the diverse population HIV-positive and negative but affected people, particularly reliability. Therefore, researchers and related practitioners may use this measure in their contexts following some validation work as needed.

## 5. Conclusions

The measurement of HIV-related stigma has the potential to help identify individuals at risk of poor psychological outcomes and inform the development and evaluation of HIV-stigma reduction interventions. However, this requires the availability of psychometrically sound measures for use among the target populations. The HSS is an example of such measures and is one of the most widely used measures of HIV-related stigma. According to this review, the HSS appears to be a reliable and valid measure of HIV-related stigma across contexts. However, evidence of validity is limited, particularly for African, Asian, Oceanian, and South American settings. This calls for more context-specific validation work of the HSS. The HSS may be used to assess HIV-related stigma following context-specific work as needed.

## Figures and Tables

**Figure 1 ijerph-18-13074-f001:**
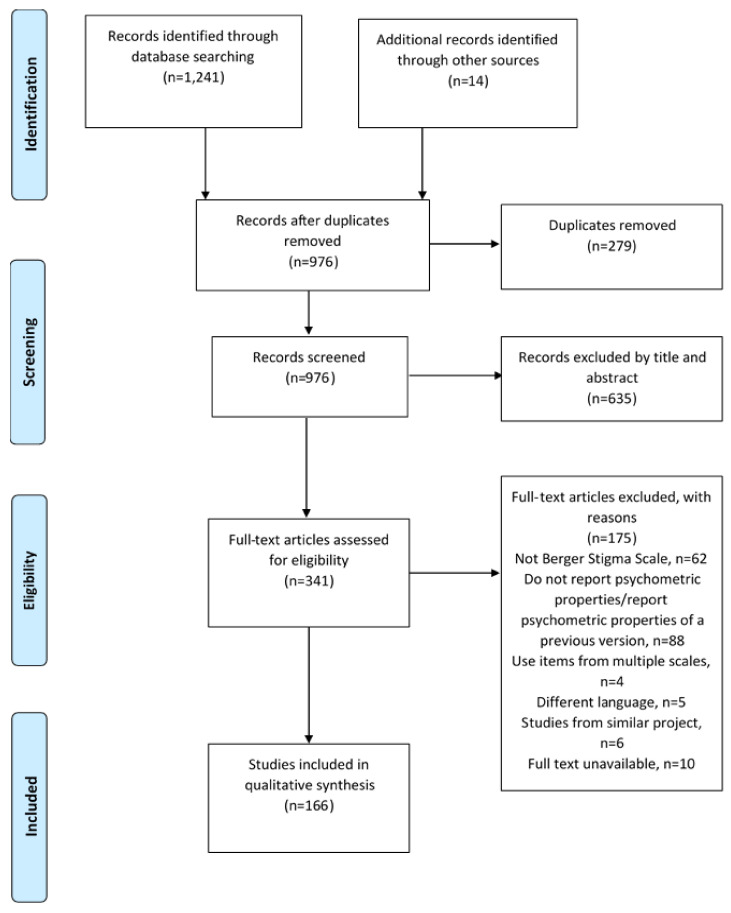
PRISMA flowchart for the systematic review process.

**Figure 2 ijerph-18-13074-f002:**
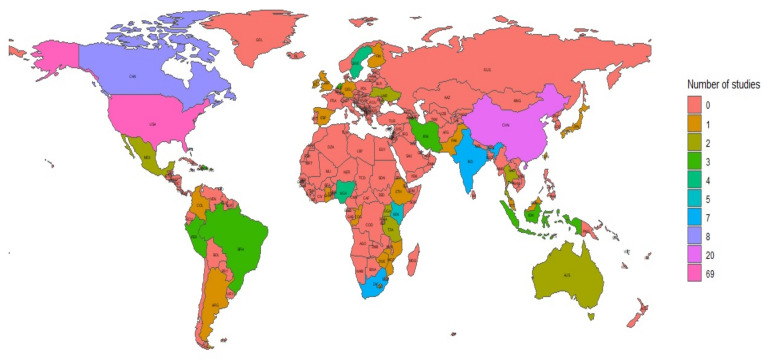
World map showing the geographic distribution of included studies.

**Table 1 ijerph-18-13074-t001:** Psychometric properties of the HSS as reported by the development and/or validation studies.

First Author (Year)	Scale Used	Reliability		Validity						
		Internal Consistency (Cronbach’s Alpha unless Otherwise Stated)	Test-Retest Reliability (icc)	Construct Validity	Convergent Validity	Divergent Validity	Discriminant Validity	Content Validity	Face Validity	Cross-Cultural Adaptation and/or Validity
Berger et al., 2001 [20]	40-item scale	0.96 for the overall scale 0.90–0.93 for the subscales	0.92 for the overall scale 0.87–0.90 for subscales	EFA revealed a 4-factor structure that reflected the dimensions of perceived HIV-related stigma and explained 46% of the total variance The four factors intercorrelated and represented one higher-order factor	Expected correlation (r) between the overall scale, including subscales, and self-esteem (−0.35–−0.68), depression (0.41–0.63), social support (−0.38–−0.54), subjective social integration (−0.50–−0.65), and social conflict (0.40–0.59) *p* < 0.001 for all correlations	NR	NR	Experts judged all the items in the scale to be clear and relevant	NR	NR
Bint-E-Saif & Shahzad, 2020 [39]	40-item scale	0.94 for the overall scale 0.81–0.91 for the subscales Split-half reliability of 0.93	0.96 for the overall scale 0.92–0.95 for the subscales	NR	Significant positive correlation (r) with depression (0.45–0.66, *p* < 0.01) Significant negative correlation (r) with self-esteem. (−0.36–−0.55, *p* < 0.01)	NR	NR	NR	NR	Scale translated to Urdu, compared and evaluated with the original version and finally back-translated to English
Boyes et al., 2010 [40]	10-item scale	0.90 for the overall scale. 0.78 and 0.88 for the subscales.	NR	EFA yielded a 2-factor structure with Eigen values of 4.87 and 1.18. The two factors intercorrelated well (r = 0.59)	Expected correlation (r) between stigma by association and depression (0.43), anxiety (0.49), bullying victimization (0.43), and peer problems (0.32) *p* < 0.01 for all correlations	NR	Stigma scores on the scale were significantly higher among HIV-affected compared to unaffected youth (*p* < 0.001)	Item relevance was determined through interviews with children, caregivers, and healthcare professionals and consultation with local academics	NR	Scale translated to Xhosa and back-translated to English by independent translators Items were then adapted to this population following in-depth interviews with participants and review by experts
Bunn et al., 2007 [45]	32-item scale	0.95 for the overall scale 0.90–0.97 for the subscales	NR	CFA retained the 4-factor structure of the original scale with good model fit (χ^2^ test *p* < 0.01; CFI = 0.908; RMSEA = 0.072)	Expected correlation (r) between the overall scale, including subscales, and self-esteem (−0.14–−0.56), stigma consciousness (0.38–0.62), discrimination (0.17–0.75), and fear of discovery (0.43–0.75) *p* < 0.005 for all correlations except for disclosure concerns subscale with self-esteem	NR	NR	NR	NR	NR
Emlet et al., 2007 [63]	40-item Scale	0.96 for the overall scale 0.92–0.96 for the subscales	NR	NR	Expected correlation (r) between the overall scale, including subscales, and depression (0.34–0.71) *p* < 0.001 or < 0.01 for all correlations except for disclosure concerns subscale with depression	NR	NR	Majority of participants felt the scale was valid and represented older people	NR	NR
Franke et al., 2010 [69]	40 and 21-item scales	0.89 and 0.84 for the overall 40 and 21-item scales, respectively. 0.68–0.89 for the subscales in the 40-item scale. 0.68–0.84 for the subscales in the 21-item scale.	NR	Factor analysis of the 21-item scale yielded a 4-factor structure with factor loadings ≥30 with no cross-loadings	The 40-item scale, including its subscales, showed expected correlations (r) with quality of life (−0.16–−0.44) and depression (0.24–0.45) *p* < 0.05 or < 0.01 for all correlations except for disclosure concerns, negative self-image, and concern with public attitudes subscales with physical health subscale in the 40-item scale The 21-item scale, including its subscales, showed expected correlations (r) with quality of life (−0.10–−0.53) and depression (0.22–0.59) *p* < 0.05 or < 0.01 for all correlations except for disclosure concerns, enacted stigma, and concern with public attitudes subscales with physical health subscale in the 21-item scale	NR	NR	NR	NR	Scale translated to Spanish and back-translated to English. A panel of experts resolved discrepancies in translation to ensure that the Peruvian-Spanish version of the scale preserved the conceptual meaning of the original scale.
Fuster-RuizdeApodaca et al., 2015 [70]	30-item scale	0.88 for the overall scale. 0.75–0.89 for the subscales.	NR	First-order CFA retained the 4-factor structure with good model fit (RMSEA = 0.054; GFI = 0.96; CFI = 0.98; χ^2^ test *p* = 0.001) Second-order CFA yielded a higher order 2-factor structure with a better model fit (RMSEA = 0.051; GFI = 0.96; CFI = 0.98)	Expected correlation (r) between the overall scale, including subscales, and social support (−0.13–−0.36), life satisfaction (−0.18–−0.34), physical and psychological well-being (−0.16–−0.30), self-efficacy to cope with stigma (−0.30–−0.54), and degree of HIV status disclosure (−0.07–−0.54) *p* < 0.01 for all correlations	NR	Perceived external stigma scores were significantly higher in those with a history of AIDS-related opportunistic infection compared to those without (*p* = 0.003)	Seven experts conducted content analysis of the interviews and coded them into relevant concepts and topics. Inter-rater reliability of the codings yielded substantial reliability (κ = 0.77, *SD* = 0.10)	A pilot sample of PLWH reviewed the final version of the scale to ensure face validity	Scale translated independently to Spanish Research team assessed the translations and reached a consensus on the final items Scale then adapted to the Spanish context following interviews with participants and expert review
Jeyaseelan et al., 2013 [91]	40 and 25-item scales	0.91 and 0.88 for the overall 40 and 25-item scales, respectively. 0.62–0.89 for the subscales in the 40-item scale. 0.19–0.88 for the subscales in the 25-item scale.	0.89 for the overall 40-item scale. 0.62–0.85 for the subscales in the 40-item scale. NR for the 25-item scale.	CFA of the 40-item scale retained the 4-factor structure but with poor model fit (RMSEA = 0.31; CFI = 0.95; CFI < 0.80 for 3 subscales) EFA and CFA of the 25-item scale retained the 4-factor structure with improved model fit (RMSEA = 0.07; CFI = 0.99 for the overall 25-item scale; RMSEA of 0.00–0.08 and CFI of 0.94–1.0 for the subscales)	Stigma scores in the 40-item scale and its subscales were significantly higher in PLWH with major depression compared to those without (*p* < 0.01) Stigma scores in the 25-item scale and its subscales were significantly higher in PLWH with major depression compared to those without (*p* < 0.05)	NR	NR	A pilot sample of participants had concerns about the repetitiveness of questions, comprehensibility of the rating scale, and the wording of items in first person Items were subsequently modified following this feedback	Overall, an expert panel rated the scale as meaningful and relevant to the Indian context Few raised concerns about the repetitiveness of the scale	Scale translated to Tamil then back-translated to English. Misinterpreted concepts clarified using the respondents’ feedback during this process.
Jimenez et al., 2010 [92]	17-item scale	0.91 for the overall scale. 0.77–0.88 for the subscales.	0.68 for the overall scale. 0.27–0.73 for the subscales.	Factor analysis replicated the 4-factor structure of the original scale with factor loadings >40	Expected correlation (r) between the overall scale, including subscales, and sexual abuse (0.15–0.29), depression (0.10–0.44), and stigmatization (0.47–0.79) *p* < 0.05 or < 0.01 for all correlations except for disclosure concerns subscale with depression and sexual abuse	NR	NR	NR	NR	Scale translated to Spanish and back-translated to English. Scale was then adapted to the Spanish population by incorporating native terms and expressions generated from FGDs.
Johnson et al., 2016 [93]	10-item scale	0.78 and 0.74 for the subscales	NR	EFA yielded a 2-factor structure accounting for 52% of the total variance with Eigen values >1; confirmed by CFA with excellent model fit statistics: χ ^2^ (34, *n* = 120) = 45.06, *p* = 0.10, *ns*, χ ^2^/*df* = 1.33, CFI = 0.96, RMSEA = 0.055	Significant positive correlations with depression (both subscales; 0.38 and 0.30, *p* < 0.01)), HIV symptoms (both subscales; 0.39 and 0.27, *p* < 0.01) Significant negative correlations with spirituality/religiousness (social stigma subscale; 0.23, *p* < 0.05), and CD4 count (self-stigma subscale; 0.32, *p* < 0.01)	NR	NR	NR	NR	NR
Kagiura et al., 2020 [95]	9-item scale	McDonald’s Omega of 0.89 for the overall scale 0.83 and 0.87 for the two subscales	NR	CFA of the 4-factor structure confirmed the original structure with sufficient model fit statistics but with insufficient reliability (two subscales with McDonald’s Omega < 0.70) EFA was then conducted, which yielded a 2-factor structure with factor loadings >0.40 and sufficient reliability	Significant positive correlation with depression (0.37–0.45, *p* < 0.001) Significant negative correlation with self-esteem. (−0.34–−0.51, *p* < 0.001)	NR	NR	NR	NR	Scale translated to Japanese and back-translated to English Both versions were then compared, and discrepancies resolved Experts then agreed that all but three items had satisfactory Japanese descriptions; these items were subsequently modified to fit the Japanese context.
Kamitani et al., 2018 [97]	10- and 13-item scales	0.90 and 0.92 for the overall 10 and 13-item scales, respectively. 0.80–0.87 for the subscales in the 10-item scale. 0.80–0.87 for the subscales in the 13-item scale.	NR	EFA yielded a 4-factor structure for the 10-item scale and a 3-factor structure for the 13-item scale. Factor loadings of 0.57–0.85 for the 13-item scale. Factor loadings of 0.70–0.86 for the 10-item scale.	Expected correlation (r) between the 13-item scale and self-reported health (−0.36, *p* = 0.003) and acculturation (−0.33, *p* = 0.006)	No association between the scale and education level (B = −0.28, *p* = 0.93)	NR	Six experts agreed that the 13-item scale had good content using the content validity index (score of 1). Items were modified based on suggestions from an initial FGD, and participants in a second FGD approved the final version to be relevant.	NR	Items were adapted to the Asian population following in-depth interviews with participants and reviews by experts.
Lindberg et al., 2014 [107]	39-item scale	0.96 for the overall scale. 0.87–0.96 for the subscales.	NR	The 4-factor structure similar to the original scale was retained on EFA. The four factors accounted for 62.2% of the variance.	Expected correlation (r) between the overall scale, including subscales, and emotional wellbeing (−0.21–−0.49, *p* < 0.01 or 0.001)	NR	NR	A pilot sample found the items to be relevant and comprehensive through think-aloud interviews	NR	Scale was translated to Swedish by three translators, then back-translated to English by an independent translator. Minor changes were made to ensure that no meaning was lost in translation.
Luz et al., 2020 [111]	12-item scale	Cronbach’s alpha, ordinal alpha and omega of 0.83, 0.88 and 0.93 for the overall scale Cronbach’s alpha of 0.69–0.87 for the subscales Ordinal alpha of 0.78–0.93 for the subscales Omega of 0.78–0.94 for the subscales	NR	A 4-factor structure showed good fit in all samples [Grindr: χ^2^ (48) = 56.9, *p* = 0.17, CFI = 0.995, TLI = 0.993, RMSEA = 0.040, SRMR = 0.062; social media: χ^2^ (48) = 63.4, *p* = 0.07, CFI = 0.993; Hornet sample χ^2^ (54) = 498.2, *p* < 0.01, CFI = 0.980, TLI = 0.973, RMSEA = 0.071, SRMR = 0.039]	NR	NR	As expected, stigma scores were significantly higher in those who were not on antiretroviral treatment compared to those on treatment for the personalized stigma in the Grindr sample (*p* = 0.02) Participants whom self-reported non-adherence scored higher than those who were adherent on personalized stigma for the 3 samples (*p* = 0.01–0.03) Among participants from the Hornet sample, those who were nonadherent scored higher with regard to concerns about public attitudes and negative self-image (*p* < 0.01)	Experts assessed the relevance of the items in the scale then a pilot sample of participants judged all items as clear, with only two items being slightly modified as needed following their feedback	NR	- Scale translated to Brazilian Portuguese and back-translated to English, evaluated by experts and pilot-tested Small DIF with respect to sample (based on the platform that they were recruited from, i.e., Grindr, Hornet, and social media) was found for one item
Mason et al., 2010 [118]	23-item scale	0.87 for the overall scale	NR	NR	NR	NR	NR	Cognitive interviews were used to assess comprehension and relevance. All participants appeared to understand the items as intended. Items were either modified or deleted following participants’ feedback.	NR	Cognitive interviews ensured cultural and developmental appropriateness of the scale.
Montano et al., 2020 [121]	7-item scale	0.73 for the overall scale 0.71 and 0.74 for the subscales	0.83 for the overall scale 0.78 and 0.79 for the subscales	Factor analysis yielded a 2-factor structure with good model fit: RMSEA = 0.038; CFI = 0.988, TLI: 0.981; SRMR = 0.05	Positive correlation between the scale, including subscales, with depression (0.26–0.53, *p* < 0.05). Negative correlation between the scale, including subscales, with adherence (−0.24–−0.31, *p* < 0.05) Negative correlation between internalized stigma subscale and CD4 levels (−0.23, *p* < 0.05) Higher levels of stigma in women (in all scores), in heterosexual men (total and enacted), in those not satisfied with their housing (total and internalized), in those previously incarcerated (total and enacted), not consistently using condoms (total and enacted), and detectable viral load (internalized) All Ps < 0.05	NR	NR	NR	NR	Scale translated to Spanish, discrepancies in translation resolved, back-translated to English, and pilot-tested
Rao et al., 2008 [144]	40-item scale	NR	NR	NR	NR	NR	NR	NR	NR	11 items showed differential item functioning across ethnic/racial groups (black vs. white participants)
Reinius et al., 2017 [147]	12-item scale	0.80–0.88 for the subscales	NR	EFA replicated the 4-factor structure of the original scale with acceptable model fit (χ^2^ = 154.2, df = 48, *p* < 0.001; CFI = 0.963; TLI = 0.93; RMSEA = 0.071)	NR	NR	NR	Experts judged that the included items best represented the different aspects of HIV stigma	NR	This study used a scale that had previously been adapted for use in this setting
Renius et al., 2018 [146]	32-item scale	NR	NR	NR	NR	NR	NR	NR	NR	Scales adapted to local contexts in the primary Indian and Swedish studies - 9 items showed DIF between cultures (Sweden, US and India), and 1 item showed DIF between males and females
Rongkavilit et al., 2010 [22]	40 and 12-item scales	0.89 and 0.75 for the overall 40 and 12-item scales, respectively. 0.77–0.93 for the subscales in the 12-item scale.	NR	EFA of the 40-item scale suggested a 5-factor structure, with four consistent with the original scale. CFA confirmed that the 12-item scale retained the 4-factor structure of the original scale with satisfactory model fit (χ^2^ test *p* > 0.5; GFI = 0.90; RMSEA = 0.02; χ ^2^/df = 1.07)	NR	NR	NR	NR	NR	Scale was forward translated to Thai then back-translated to English by independent translators
Valle et al., 2015 [171]	21-item scale	0.88 for the overall scale. 0.76–0.83 for the subscales.	NR	EFA extracted a 4-factor structure with adequate model fit (χ^2^ (210, *n* = 75) = 752.413, *p* < 0.001; KMO = 0.777)	Expected correlation (r) between the overall scale, including subscales, and measures of depression (0.25–0.34, *p* < 0.01) and self-esteem (significant negative correlation)	NR	Significant differences between the high score and low score groups (*p* < 0.01 for all subscales)	NR	NR	NR
Wiklander et al., 2013 [23]	8-item scale	0.81 for the overall scale. 0.55–0.80 for the subscales	NR	PCA yielded a 3-factor structure with all items loading on the same factors similar to the original scale Moderate to high correlations between the scale and its subscales. Low correlation between the subscales	Expected correlation (r) between the overall scale, including subscales, and quality of life (−0.26–−0.49) *p* < 0.05, <0.01, or <0.001 for all correlations except for negative self-image subscale with social inclusion subscale	NR	NR	NR	NR	A previously adapted Swedish version of the scale was adapted to children by paediatric experts.
Wright et al., 2007 [178]	10-item scale	0.72–0.84 for the subscales.	NR	4-factor structure with high correlations to the original scale	Expected correlation (r) between the overall scale, including subscales, and depression (0.35–0.41), anxiety (0.24–0.32, social support (−0.20–−0.35), and alcohol use (0.24–0.35) *p* < 0.05 or <0.01 for all correlations except for total stigma with anxiety and negative self-image with social support and alcohol use	NR	NR	NR	NR	NR
Yu et al., 2019 [184]	18-item scale	0.92 for the overall scale. 0.73–0.91 for the subscales.	NR	CFA confirmed a 4-factor structure similar to the original scale with a good model fit (χ^2^ = 11399.49, *p* < 0.01; GFI = 0.94; AGFI = 0.91; RMSEA = 0.55; AIC = 420.34)	Expected correlation (r) between the overall scale, including subscales, and depression (0.25–0.46, *p* = 0.01)	NR	NR	Five experts judged the items and the scale to have adequate content validity in terms of relevance, cultural equivalence, and clarity (both item and scales content validity index score of 1)	NR	Scale was translated to Chinese then back-translated to English by three independent experts. Items then adapted to this population following group discussions involving experts. Further modifications were made following a pilot test and a review of the tool by some participants.

PCA—principal component analysis, CFA—confirmatory factor analysis; EFA—exploratory factor analysis; CFI—comparative fit index; DIF—differential item functioning; GFI—goodness of fit index; AGFI—adjusted goodness of fit index; AIC—Akaike information criterion; χ2—Chi-square; df-—degree of freedom; RMSEA—root mean square of approximation; SRMR—standardized root mean square residual; TLI—Tucker–Lewis index; a—Cronbach alpha; ICC—intraclass correlation; r—Pearson correlation coefficient; KMO—Kaiser–Meyer–Olkin measure; FGDs—focused group discussions; PLWH—people living with HIV; NR—not reported.

## Data Availability

The data supporting the conclusions presented in this article are available within this article and the Appendix A.

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
