# Peer review of "Psychometric Properties of the Berger HIV Stigma Scale: A Systematic Review"

_ijerph, 2021, doi:10.3390/ijerph182413074_

Round 1

Reviewer 1 Report

Minor Comments:

  1. Please change title to: “Psychometric properties of the Berger HIV Stigma Scale: A Systematic Review”
  2. Please include a brief epidemiological description of the HIV situation around the world to support the relevance of having validated scales to assess HIV stigma.
  3. Please include some thoughts in the discussion on why the majority of studies were conducted in North America and Europe, and what can be done to change that.
  4. Please further develop de limitations section, according to Shea BJ, Grimshaw JM, Wells GA, Boers M, Andersson N, Hamel C, et al. Development of AMSTAR: a measurement tool to assess the methodological quality of systematic reviews. BMC Med Res Methodol. 2007;7:10

Best wishes.

Reviewer 2 Report

The paper presents a rigorous systematic review about the psychometric properties of the HSS scale proposed in Berger et al. (2001). The methodological aspects of the selection of articles are sufficiently detailed and are consistent with the standard criteria used in this topic.

Minor Points:

1. For each selected paper, comprehensive information on the psychometric properties evaluated is provided. At this point, to facilitate the reading of the paper, I could suggest that the authors keep Table 1 (focused on the validation articles of the HSS scale) in the body of the papers and include Table 2 (relative to the experimental articles) as supplementary material, since most only deal with reliability and the information is perfectly summarized in the text.

2. The discussion and conclusions sections are consistent with the results of the paper and provide relevant information for the use of the HSS scale, highlighting the relative scarcity of evidence about some aspects of the validity of the scale.

3. Given that there are studies in a range of 20 years (2001-2021) and that the perception and incidence of HIV may have changed, I do not know if it would be possible for the authors to include any information or reflection on the apparent stability over the 20 years of the psychometric properties of the scale between the initial and the most recent articles, or if a slight change is detected?
